

# Molecular insights into programmed cell death in esophageal squamous cell carcinoma

Min Chen[1,2,3,4,5,6,7], Yijun Qi[2,3,4,5,6,7], Shenghua Zhang[2,3,4,5,6,7], Yubo Du[2,3,4,5,6,7], Haodong Cheng[2,3,4,5,6,7] and Shegan Gao[1,2,3,4,5,6,7]

[1] School of Information Engineering, Henan University of Science and Technology, Luoyang, China
[2] State Key Laboratory of Esophageal Cancer Prevention and Treatment, The First Affiliated Hospital of Henan University of Science and Technology, Luoyang, China
[3] Henan Key Laboratory of Microbiome and Esophageal Cancer Prevention and Treatment, The First Affiliated Hospital of Henan University of Science and Technology, Luoyang, China
[4] Henan Key Laboratory of Cancer Epigenetics, The First Affiliated Hospital of Henan University of Science and Technology, Luoyang, China
[5] Cancer Hospital, The First Affiliated Hospital of Henan University of Science and Technology, Luoyang, China
[6] College of Clinical Medicine, Henan University of Science and Technology, Luoyang, China
[7] Medical College of Henan University of Science and Technology, Luoyang, China

## ABSTRACT

**Background**. Esophageal squamous cell carcinoma (ESCC) is a deadly type of esophageal cancer. Programmed cell death (PCD) is an important pathway of cellular self-extermination and is closely involved in cancer progression. A detailed study of its mechanism may contribute to ESCC treatment.

**Methods**. We obtained expression profiling data of ESCC patients from public databases and genes related to 12 types of PCD from previous studies. Hub genes in ESCC were screened from PCD-related genes applying differential expression analysis, machine learning analysis, linear support vector machine (SVM), random forest and Least Absolute Shrinkage and Selection Operator (LASSO) regression analysis. In addition, based on the HTFtarget and TargetScan databases, transcription factors (TFs) and miRNAs interacting with the hub genes were selected. The relationship between hub genes and immune cells were analyzed using the CIBERSORT algorithm. Finally, to verify the potential impact of the screened hub genes on ESCC occurrence and development, a series of *in vitro* cell experiments were conducted.

**Results**. We screened 149 PCD-related DEGs, of which five DEGs (*INHBA*, *LRRK2*, *HSP90AA1*, *HSPB8*, and *EIF2AK2*) were identified as the hub genes of ESCC. The area under the curve (AUC) of receiver operating characteristic (ROC) curve of the integrated model developed using the hub genes reached 0.997, showing a noticeably high diagnostic accuracy. The number of TFs and miRNAs regulating hub genes was 105 and 22, respectively. *INHBA*, *HSP90AA1* and *EIF2AK2* were overexpressed in cancer tissues and cells of ESCC. Notably, *INHBA* knockdown suppressed ECSS cell migration and invasion and altered the expression of important apoptotic and survival proteins.

**Conclusion**. This study identified significant molecules with promising accuracy for the diagnosis of ESCC, which may provide a new perspective and experimental basis for ESCC research.

Corresponding author
Shegan Gao, gsg112258@163.com

## INTRODUCTION

In 2020, the world had around 604,000 new cases of esophageal cancer and 544,000 related deaths. It is expected that annual new case and death of esophageal cancer will increase to 957,000 and 880,000 by 2040, respectively (*Morgan et al., 2022*; *Zhan et al., 2022*). Adenocarcinoma and squamous cell carcinoma are two major histological subtypes of esophageal cancer (*Abnet, Arnold & Wei, 2018*). Patients with localized esophageal squamous cell carcinoma (ESCC) are normally treated by endoscopic resection or surgery, while those metastatic ESCC are suitable for taking chemoradiotherapy (*Codipilly & Wang, 2022*). ESCC often has a dismal prognosis, especially for those diagnosed with advanced cancer, due to incomplete understanding of the mechanisms of ESCC and a lack of effective cancer diagnosis, patient prognosis, and treatment (*Lee et al., 2018*). In recent years, the identification of biomarkers has improved the accuracy and availability of early screening for ESCC (*Li et al., 2020*; *Tsoneva et al., 2023*). To elucidate the molecular mechanisms of ESCC and help select targets for early treatment and diagnosis, some researchers screened hub genes by integrating multiple databases and applying bioinformatics tools (*Song et al., 2021*; *Yang et al., 2019*). However, the etiology and molecular mechanisms of ESCC are still unclear, and new biomarkers and therapeutic candidate targets for ESCC should be discovered. Programmed cell death (PCD) is a regulatory cell death and an important terminal pathway that requires energy (*Sun & Peng, 2009*; *Tower, 2015*). Stressed, damaged, malignant, or infected cells are lysed and efficiently eliminated *via* PCD (*Liu et al., 2022*). Currently known types of PCD include, but are not limited to, apoptosis and necroptosis, pyroptosis, autophagy-dependent cell death, cuproptosis, netotic cell death, ferroptosis, entotic cell death, lysosome-dependent cell death, parthanatos, alkaliptosis, and oxeiptosis (*Peng et al., 2022*; *Dong et al., 2024*; *Kim, Kim & Lee, 2023*). Aberrant regulation of PCD is considered a key feature of carcinogenesis (*Fuchs & Steller, 2011*). PCD disorder is a major molecular mechanism of each subroutine that could provide a range of possible targets for cancer diagnosis and therapy (*Ke et al., 2016*). However, in the treatment of ESCC patients, the specific roles of distinct regulated cell death (RCD) subroutines vary in different patients and they may jointly determine the fate of cancer cells (*Mishra et al., 2018*). Therefore, the simultaneous manipulation of multiple RCD signaling pathways by dual-target or multi-target small molecules has greater potential in cancer therapy (*Zhang et al., 2023*).

Artificial intelligence (AI) is a promising approach to reveal the mechanisms of ESCC development and potential biomarkers by analyzing large amounts of data and identifying complex patterns (*Zhang et al., 2023*). The present research employed three machine learning methods to analyze 12 PCD-related genes, from which we identified the diagnostic genes for ESCC and provided information for understanding their tumor immunology and molecular mechanisms in ESCC.

## MATERIALS AND METHODS

PCD-related genes in 12 models were obtained from the article published by *Zou et al. (2022)*, including necroptosis, apoptosis, cuproptosis, pyroptosis, alkaliptosis, ferroptosis, parthanatos, netotic cell death, autophagy-dependent cell death, lysosome-dependent cell death, oxeiptosis, and entotic cell death (Table S1).

All data were retrieved from GSE53625 and GSE43732 (Gene Expression Omnibus, GEO, https://www.ncbi.nlm.nih.gov/geo/). GSE53625 contained 179 normal samples and 179 ESCC samples. The sequence information and clinicopathological data of GPL18109 chip platform were obtained from GSE53625 dataset (*Gao et al., 2021*). For the initial GPL18109 microarray platform sequence information of the dataset, the probe sequences were re-annotated according to the latest alignment file in the GENCODE database (https://www.gencodegenes.org/) (*Frankish et al., 2021*), and when multiple probes matched to a gene, the mean value was calculated as the value of gene expression. The miRNA matrix data files and clinicopathological data of 119 ESCC samples and normal samples were obtained from GSE43732 dataset (*Chen et al., 2014*).

We obtained gene expression data and relevant clinical information of esophageal cancer patients from The Cancer Genome Atlas (TCGA, https://portal.gdc.cancer.gov/) database to validate the screened hub genes (*The Cancer Genome Atlas Research Network, 2017*). FPKM values of gene expression data in TCGA were converted to transcripts per kilobase million (TPM) and log2 transformed. ESCC samples were ultimately retained based on clinical information, and a total of 80 ESCC samples and 11 normal control samples were included.

### Screening of differentially expressed genes

The R package limma (version 3.42.2) (*Ritchie et al., 2015*) was used to analyze the differential expression between the control and ESCC groups of GSE53625 dataset, with 179 samples in each group, and the threshold for the selection of differentially expressed genes (DEGs) was adj. $P < 0.05$ and $|\log_2 FC| > 1$. Then the difference analysis results were visualized into volcano map and heat map.

### Functional enrichment analysis

The selected DEGs were the input to the R package clusterProfiler (version 0.4.6) (*Wu et al., 2021*), and the internal enrichGO and enrichKEGG functions were used for GO and KEGG functional enrichment analysis after reading the gene list file. GO was adopted to explore the functional changes in terms of cell component (CC), molecular function (MF), and biological process (BP) caused by DEGs. The effects on pathways were investigated using KEGG analysis.

### Machine learning analysis

Linear support vector machine (SVM), random-forest and Least Absolute Shrinkage and Selection Operator (LASSO) regression analysis methods were the three machine learning classifiers used for feature selection. The selection of genes for the Linear SVM method was conducted using the "rfe" function in the R and run with 100-fold cross validation. As a commonly used genomic data analysis, random forest is a classification and regression

trees (CART) ensemble method that is trained on imported samples and randomly selected features (*Liu & Zhao, 2017*). Decision trees have the advantages of easy use and interpretation, outlier resistance, efficient handling of numerous predictor variables, and built-in mechanism for processing missing data with relevant variables (*Alderden et al., 2018*). To establish a random forest model, we employed the R package RandomForest (version 4.6–14) (*Liaw & Wiener, 2015*). The number of variables for binary trees in a specified node of the model (mtry) was determined according to the average model error rate of out of bag (OOB) samples, while the number of decision trees contained in the random forest (ntree) was determined according to the relationship between model error and the number of decision trees. Based on the identified variables and the ranking of important variables, a random forest model was established. LASSO regression analysis was conducted in the R package ''glmnet''(version 4.1-2) (*Engebretsen & Bohlin, 2019*), the cross-validated parameter was nfolds =10 and other parameter was set to family ='binomial'.

## Identification and performance evaluation of hub genes

Genes filtered by linear SVM, RandomForest and LASSO regression analysis were compared, and those jointly selected by the three machines were defined as the hub genes. The ''pROC'' package (*Robin et al., 2011*) served to evaluate the specificity and sensitivity of genes for ESCC diagnosis by generating ROC curves for each gene and calculating the AUC.

## Molecular regulatory network analysis of the hub genes

The regulatory correlations between human TFs and their target genes were collected from the HTFtarget database (http://bioinfo.life.hust.edu.cn/hTFtarget) (*Zhang et al., 2020*), and the interactions between human miRNAs and their target genes were obtained from the TargetScan database (http://www.targetscan.org) (*Agarwal et al., 2015*). Using these two databases, we screened TFs and miRNAs that interacted with the hub genes, and then cytoscape (http://cytoscape.org/, version 3.7.2) (*Shannon et al., 2003*) was employed to draw the regulatory network for TFs-hub genes and miRNAs-hub genes.

## Immune infiltration analysis

CIBERSORT is an immune infiltration analysis method on the basis of linear support vector regression for deconvolution analysis (*Newman et al., 2015*). There are two files for CIBERSORT (https://cibersort.stanford.edu/) input, one is the sequencing expression matrix, the rows of which represent the expression value of a given gene, and the other text file is the ''signature matrix''. Based on the score of obtained immune cell infiltration in each sample, the correlation of immune cell infiltration and hub gene expression in a sample could be explored using Spearman correlation analysis.

## Acquisition of cells and transfection of siRNA

Human ESCC cell line KYSE-410 (product number: BNCC359845) and normal human esophageal squamous epithelial cell line Het-1A (product number: BNCC342346) were purchased from the official website of BNCC (Beijing, China). DMEM containing 10% fetal

bovine serum (FBS, Gibco, Billings, MT, USA), penicillin (100 U/mL) and streptomycin (100 $\mu$g/mL) (HyClone, USA) was used to culture all the cells and incubated together with 5% $CO_2$ at 37 °C. GenePharma (Shanghai, China) synthesized *INHBA*- targeted siRNA (si-*INHBA*). According to the manufacturer's transfection protocol, transfection operations were performed using INTERFERin transfection reagent (Polyplus-transfection SA, France). When the cells grew to 50–70% fusion, transfection was conducted in antibiotic-free medium, which was replaced with fresh FBS-containing medium after 48h of transfection. Next, the culture was continued for subsequent experiments. The si*INHBA* sequence (5′-CCAACAGGACCAGGACCAA-3′) and si-NC sequence (5′-UUCUCCGAACGUGUCACGU-3′) were synthesized by Sangon, Shanghai, China.

## Quantitative reverse transcription PCR

Total RNA was extracted by adding Trizol (Invitrogen, USA) reagent and the concentration and purity were determined by ultraviolet spectrophotometry (Thermo Fisher Scientific, Waltham, MA, USA). RNA was reverse-transcribed into cDNA using NovoScript All-in-one SuperMix (Novoprotein, China), and the results were analyzed on the 7500 Fast Real-Time system (AB, USA) for qRT-PCR. Primer design was performed by Sangon Biotech (Shanghai, China) and synthesized against *EIF2AK2*, *HSP90AA1*, and *INHBA* genes. GAPDH was an internal reference gene. PCR reactions were performed with PowerUp™ SYBR™ Green Master Mix (Applied Biosystems, Waltham, MA, USA), according to the manufacturer's protocol, which consisted of a pre-denaturation step at 95 °C for 2 min (min), followed by 40 cycles, with each cycle consisting of denaturation at 95 °C for 15 s (s) and extension at 60 °C for 60 s. The final data were analyzed for relative quantification using the $2^{-\Delta\Delta Ct}$ method (*Livak & Schmittgen, 2001*). The primer sequence used was synthesized by Sangon, Shanghai, China (Table 1).

## Transwell assay

To carry out migration and invasion assays, cell culture inserts (8 $\mu$M pore size, BD, USA) and Matrigel invasion chambers (BD, Franklin Lakes, NJ, USA) were used, respectively. After 24 h of si-*INHBA* interference on KYSE-410 cells, the cells were starved (cultured for 24 h using serum-free medium) to synchronize the cell state. After the transfection of serum-starved ESCC cells, approximately $2 \times 10^4$ cells were inoculated into the upper chamber, while the lower chamber contained medium dissolved with 10% FBS. The cells were stained with a concentration of 0.2% crystal violet and observed from four randomly selected areas under a Zeiss AX10 inverted microscope (Carl Zeiss, Oberkochen, Germany) to capture images.

## Western blot assay

To detect the expression of cleaved caspase3, cell cycle proteins Bax and Bcl-2, we extracted total proteins from the logarithmic growth phase human ESCC cell line KYSE-410 using cellular protein extraction reagents. The BCA method was used to determine the protein content. The protein was heated at 95 °C for 10 min, followed by loading 20 $\mu$g of the sample onto a polyacrylamide gel for electrophoresis. Subsequently, the protein isolated from the gel was moved onto the PVDF membrane. After being blocked with 5% skimmed milk

**Table 1  Primer sequences for qRT-PCR.**

| Gene | Forward primer sequence (5′–3′) | Reverse primer sequence (5′–3′) |
|---|---|---|
| EIF2AK2 | GCCGCTAAACTTGCATATCTTCA | TCACACGTAGTAGCAAAAGAACC |
| HSP90AA1 | CATAACGATGATGAGCAGTACGC | GACCCATAGGTTCACCTGTGT |
| INHBA | CAACAGGACCAGGACCAAAGT | GAGAGCAACAGTTCACTCCTC |
| GAPDH | CTGGGCTACACTGAGCACC | AAGTGGTCGTTGAGGGCAATG |

for 2 h, the membrane was incubated with anti-caspase-3 (9662, 1:1000, Cell Signaling Technology, Danvers, MA, USA), anti-Bax (2772, 1:1000, Cell Signaling Technology, Danvers, MA, USA), anti-Bcl-2 (15071, 1:1000, Cell Signaling Technology, Danvers, MA, USA), anti-GAPDH (sc-137179, 1:1000, Santa Cruz Biotechnology, Dallas, TX, USA) for 24 h at 4 °C. Next, Bio-Rad's horseradish peroxidase-labeled secondary antibody (ab6721, 1:5000, Abcam, Cambridge, UK) was added for further incubation at room temperature for 1 h. Then SuperSignal West Pico chemiluminescent substrate from Thermo Fisher Scientific was used to treat the membrane and signal were detection using the Bio-Rad Chemidoc XRS+ system. Data were quantified with Bio-Rad Image Lab software and normalized to GAPDH.

## Statistical analysis

All analyzed bioinformatics data were imported into R software (version 4.0.5) for analysis. All the experiments were conducted at least three times. The normality of the data distribution was assessed using the Shapiro–Wilk test. Student's $t$-test and Wilcoxon rank sum test were used to compare differences between tumor and normal control samples and between ESCC cell lines and their normal controls, where data were expressed as mean ± standard error of the mean (SEM). Venn diagrams were plotted using the R "VennDiagram" package (*Chen & Boutros, 2011*). When defining the statistical significance, the default $p$ value <0.05 was considered statistically significant.

## RESULTS

### Functional characterization of DEGs between ESCC and normal samples

By analyzing the difference of mRNA expression profile data between ESCC samples and normal samples in GSE53625 (Fig. 1A), 2,606 DEGs were identified and 149 of them were PCD-related genes. The expression heatmap of the 50 PCD-related DEGs with the largest fold change in normal tissues and ESCC cancer tissues was shown in Fig. 1B. The 149 PCD-related DEGs significantly annotated by the necroptosis, ferroptosis and immunomodulatory pathways (lysosome, tuberculosis, *IL–17* signaling pathway, rheumatoid arthritis, cytokine–cytokine receptor interaction, Influenza A) were selected by performing the KEGG pathway annotation analysis (Fig. 1C). The BPs enriched with 149 PCD-related DEGs also all mediated the regulation of PCD (Fig. 1D). From the annotated CC and MF terms, 149 PCD-related DEGs were significantly associated with extracellular matrix, lumen related to secretory structure and protein transport, and secreted small molecules, including cytokines and growth factors (Figs. 1E–1F).

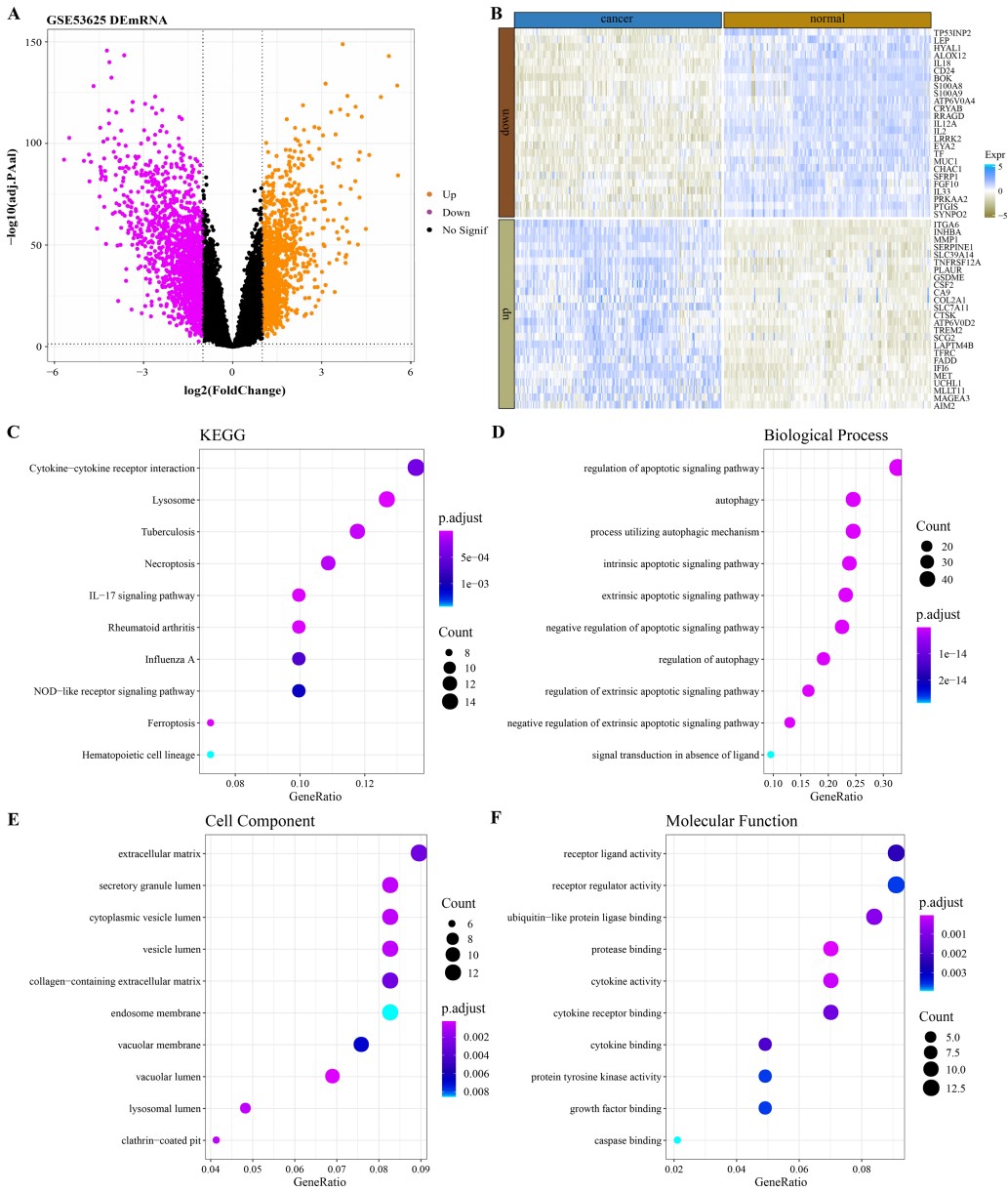

**Figure 1 Functional characterization of DEGs between ESCC and normal samples.** (A) Based on the GSE53625 dataset, and 2606 differentially expressed genes were screened by the limma package (screening queues of adj. $p < 0.05$ and $|log_2FC| > 1$). (B) Expression heatmap of 50 PCD-related DEGs with the largest Fold change in ESCC cancer tissues and normal tissues. (C) Top 10 KEGG pathways annotated by 149 PCD-related DEGs. (D–F) Top 10 GO BPs, GO CC and GO MF enriched by 149 PCD-related DEGs.

## Screening of PCD-related DEGs by machine learning algorithms

We used linear SVM to assess the performance of classifiers composed of different gene combinations of 149 PCD-related DEGs and determine a classifier consisting of nine PCD-related DEGs with the highest accuracy (Fig. 2A). Machine learning based on LASSO regression analysis selected genes corresponding to the smallest binomial deviance, here,

Peer J

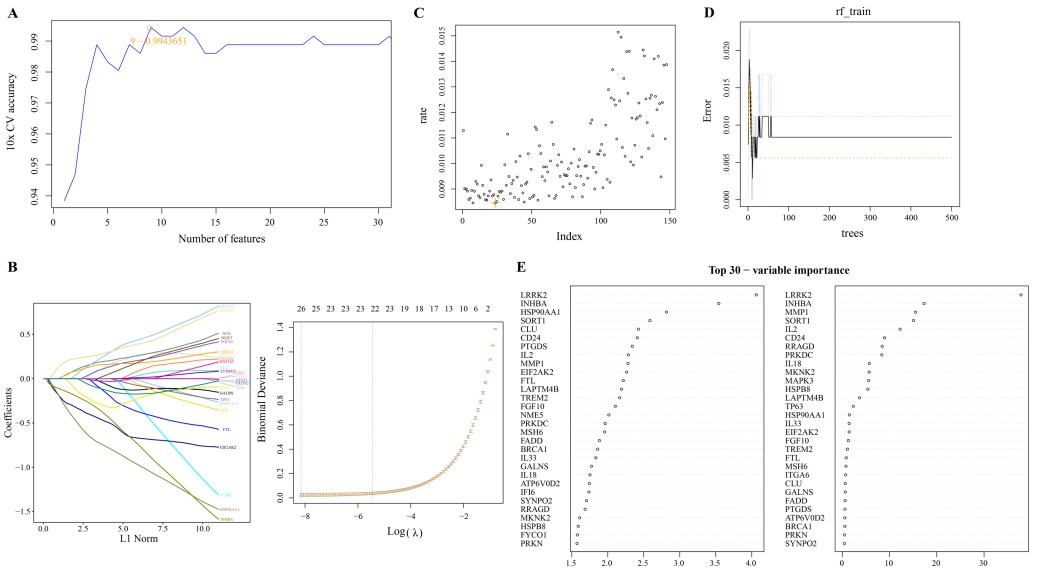

**Figure 2** Screening of PCD related DEGs by machine learning algorithms. (A) Accuracy corresponding to the number of genes in the Linear SVM classifier. (B) LASSO regression analysis. (C) The mtry determined based on the mean error rate of the model for OOB samples. (D) Relationship between the number of model error and decision tree. (E) Importance ranking of 30 PCD-related DEGs based on mean decrease accuracy and mean decrease Gini output.

26 of 149 PCD-related DEGs were filtered (Fig. 2B). Based on the mean misjudgment rate of the model for the OOB samples, we determined mtry = 24 (Fig. 2C). Then, according to the correlation between the model error and the number of decision trees, the ntree was determined to be 50, after which the estimated OOB error rate was quite stable (Fig. 2D). The results of the random forest analysis output were arranged according to the importance of the PCD-related DEGs in the ranking order, and the top 30 genes were shown (Fig. 2E).

## Identification of hub genes and evaluation of their diagnostic efficacy for ESCC

A total of five overlapping genes of PCD-related DEGs were jointly selected by the three machine learning algorithms and they were considered as hub genes of ESCC (Fig. 3A). Five hub genes were synthesized into a comprehensive model and the AUC of the ROC curve of the model reached 0.997, meaning that the diagnostic specificity and sensitivity of the model were high (Fig. 3B). All the hub genes were also differentially expressed between ESCC cancer tissues and normal tissues. Particularly, the expression of *INHBA*, *HSP90AA1* and *EIF2AK2* in ESCC cancer tissues was significantly higher than that in normal tissues, while the level of *LRRK2* and *HSPB8* in ESCC tissues was remarkably lower as compared to the normal tissues (Fig. 3C). The expression level of the five hub genes and the diagnostic effect of each hub gene was validated in the TCGA-ESCA dataset. Among them, INHBA, HSP90AA1, and EIF2AK2 were significantly higher expressed in tumor samples than in normal samples (Fig. S1A). We found that the validated model of INHBA had an AUC value of 0.927 for the ROC curve, HSP90AA1 had an AUC value of 0.934, and ELF2AK2

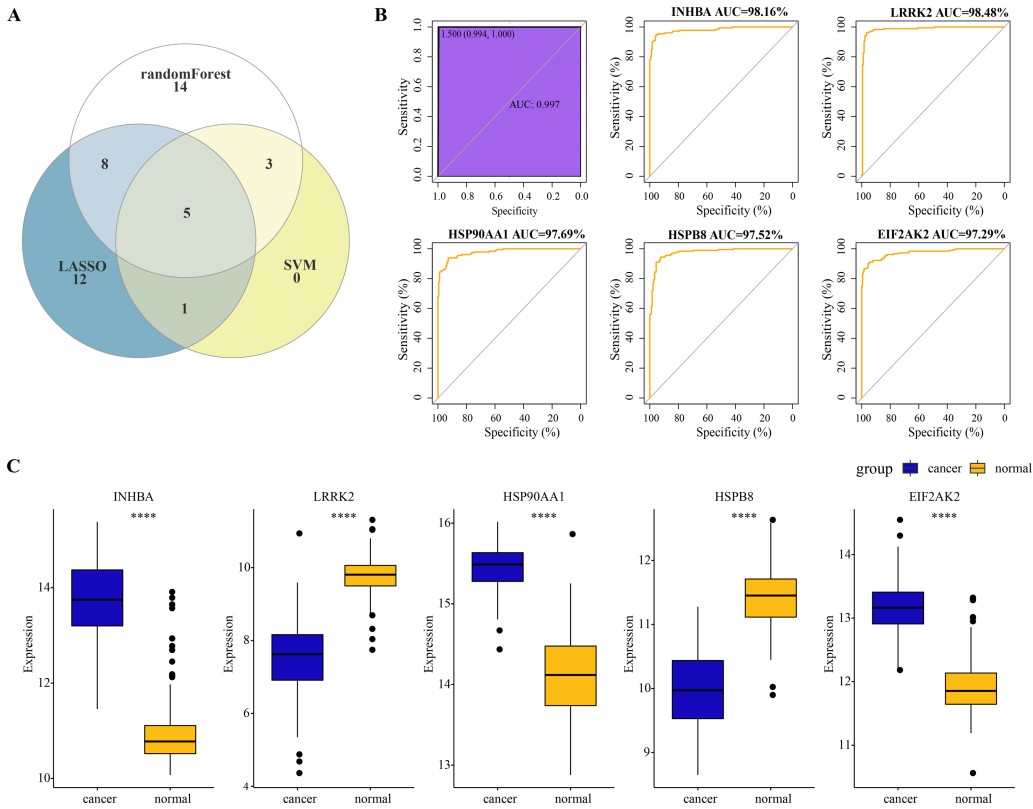

**Figure 3** **Identification of hub genes and evaluation of their diagnostic efficacy for ESCC.** (A) The Venn diagram shows the overlapping genes of PCD-related DEGs selected by three machine learning algorithms. (B) Diagnostic ROC curves for comprehensive models fitted for the five hub genes and each hub. (C) Wilcoxon rank sum test to be used to compare the difference in expression levels of the five hub genes between ESCC tissues and normal tissues. **** represents $p < 0.0001$. Data were presented as median.

had an AUC value of 0.871, whereas the LRRK2 and HSPB8 models had lower AUC values of the ROC curve (0.517 and 0.536, respectively) (Fig. S1B).These results suggest that the hub genes we identified have a high potential for diagnostic efficacy in ESCC.

## Upstream regulators of diagnostic hub genes in ESCC

Differential miRNA expression analysis between ESCC tissues and normal tissues in GSE43732 cell line revealed 136 differentially expressed miRNAs. According to the regulatory relationship between TF and its target genes provided by HTFtarget database and the interaction between human miRNA and target genes predicted by TargetScan database, the five hub genes of ESCC were regulated by 22 miRNAs, which also showed potential indirect interactions with 105 potential TFs regulating the five hub genes (Fig. 4).

## Prognosis and immune relevance of the hub gene

PCA revealed that immune cells had significant group-biased clustering and individual differences between ESCC tumor tissues and normal tissues (Fig. 5A). CD8 T cells were

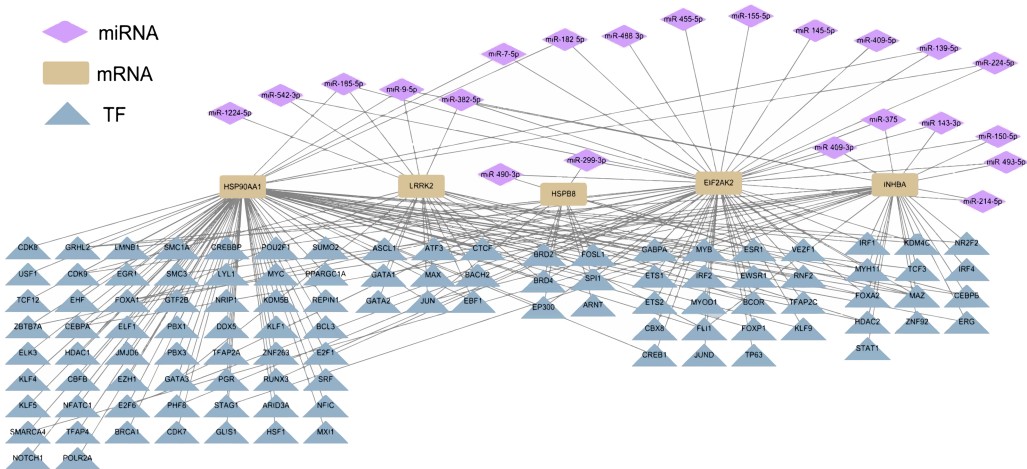

**Figure 4** Potential TF-hub gene-miRNA regulatory network of diagnostic hub genes in ESCC.

the main executors of anti-tumor immune responses. The high correlation of CD8 T cells with monocytes, activated mast cells, naive CD4 T cells, M0 macrophages, resting CD4 T memory cells, activated CD4 memory cells, follicular helper T cells, Tregs and M1 macrophages suggests their interplay in the tumor microenvironment (Fig. 5B). Compared with the normal tissue of ESCC, the proportion of naive CD4 T cells, memory B cells, M0 and M1 macrophage, activated CD4 T memory cells in tumor tissue increased significantly, while the proportion of gamma delta T cells, CD8 T cells, Tregs, monocytes, activated NK cells, resting CD4 T memory cells, resting and activated mast cells, and M2 macrophage significantly reduced (Fig. 5C). As shown in Fig. 5D, *INHBA* correlated strongly with macrophage M0 as well as T cells regulatory, with a Spearman correlation coefficient close to 0.5. *LRRK2* was weakly correlated with mast cells resting, *HSP90AA1* with T cells CD4 naive, *HSPB8* with macrophage M2 and mast cells resting, and *EIF2AK2* with macrophage M1. Although these five hub genes are weakly correlated with immune cells, they still provide some clues to the immunomodulatory role of these genes in ESCC. This further implies that the effects of these hub genes in the ESCC tumor microenvironment may be cell type specific. These results suggested that these hub genes had a potential impact on the immune microenvironment in ESCC. Subsequently, the ESCC samples from the GSE53625 dataset were divided into high and low expression groups according to the optimal cutoff value of each gene expression. As shown in Fig. S2 , we observed that patients with low expression of *INHBA*, *HSP900AA1*, *HSPB8*, and *EIF2AK2* had a significantly better prognosis than the high expression group.

## Expression of the hub genes in ESCC cells

Human ESCC cell line KYSE-410 were used as experimental cells and human esophageal epithelial cell Het-1A as control cells to analyze the mRNA levels of *EIF2AK2*, *HSP90AA1* and *INHBA*. The mRNA levels of all these three molecules were significantly higher in KYSE-410 cells than those in Het-1A cells (Figs. 6A–6C). After knockdown of *INHBA* expression in KYSE-410 cells, cell migration and invasion were significantly suppressed

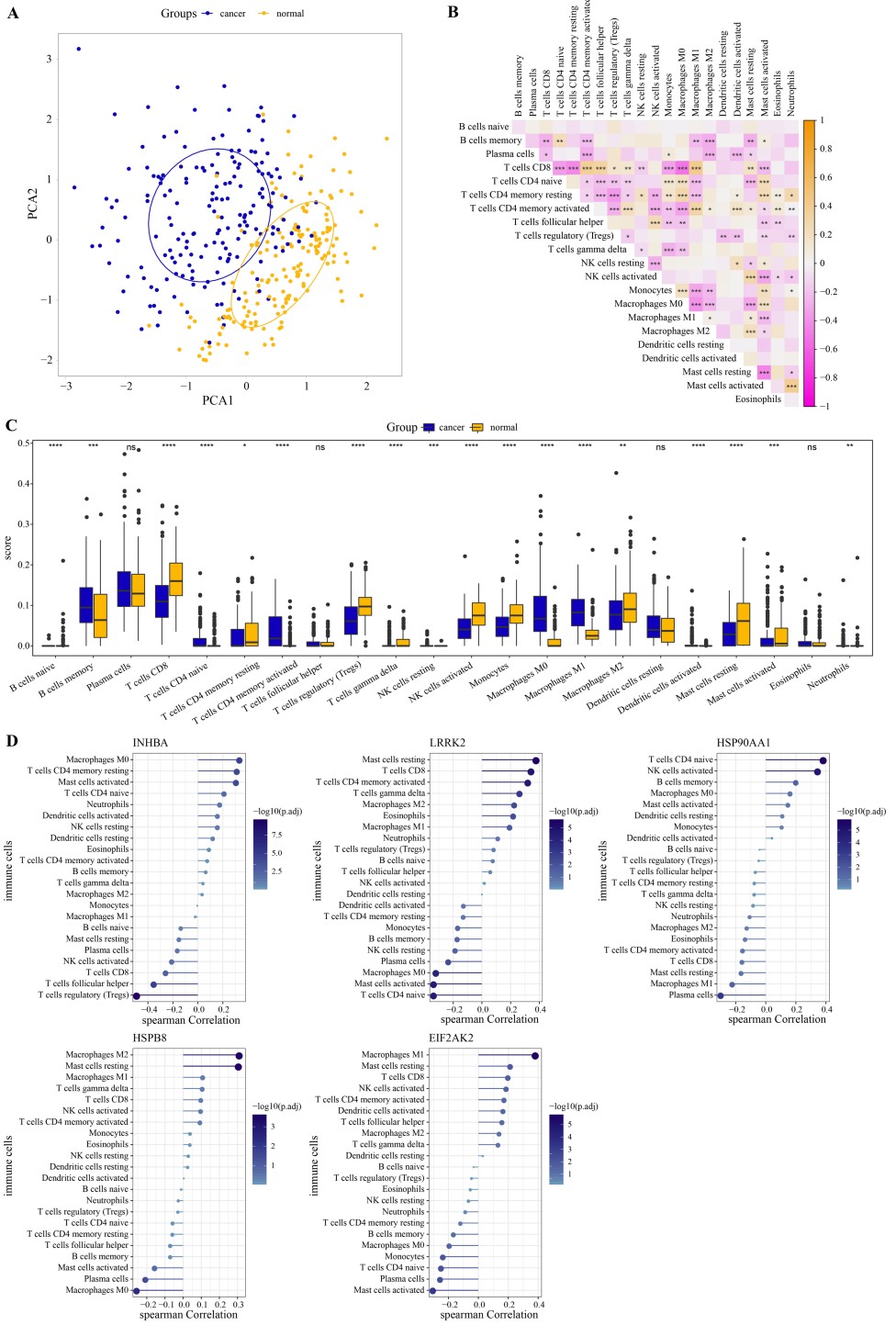

**Figure 5 Immune correlation of hub genes.** (A) PCA cluster diagram of immune cell infiltration between tumor samples and control samples. (B) Heatmap of 

**Figure 5 (…continued)**
correlations between 22 immune cells. (C) The CIBERSORT algorithm was used to assess the extent of immune cell infiltration between samples. The differences in the proportion of immune cell infiltration between ESCC and normal samples were compared by the Wilcoxon rank sum test. (D) The Spearman correlation analysis between each hub gene and the degree of immune cell infiltration. * represents $p < 0.05$, ** represents $p < 0.01$, *** represents $p < 0.001$, **** represents $p < 0.0001$, and ns represents no difference. Data were presented as median.

(Figs. 6D, 6E). Thus, *INHBA* played a crucial role in the migration and invasion of ESCC cells. Subsequently, to further analyze the role of *INHBA* in regulating the survival and death of ESCC cells, we measured the expression of Bcl-2, caspase-3, and Bax using Western blot analysis. A significant upregulation of Caspase-3 and Bax was observed after silencing the expression of *INHBA* as compared to the control group, with a significantly lower expression of Bcl-2 than in the si-NC group (Fig. 7). This revealed that gene silencing by *INHBA* can alter the expression of apoptotic and survival proteins and may drive the PCD of ESCC cancer cells. These results demonstrated that the hub genes we identified might have a potential impact on the development of ESCC.

## DISCUSSION

PCD escape is one of several critical early events during the transformation from normal cellular homeostasis to malignancy (*Chao, Majeti & Weissman, 2011*), suggesting that certain key effector molecules of PCD may have a strong potential in early diagnosis of cancer (*Gong et al., 2023*). Research showed that different types of PCD share a coordinated system (*Liu et al., 2022*). Based on these findings, if only targeting a single pathway of regulated cell death (RCD) without considering other forms of RCD as a redundant fatal backup function, the treatment may eventually fail because the boundaries between various pathways of cell death are blur at the molecular level (*Gong et al., 2023*). Using single-cell RNA transcriptome data, some researchers developed a 16-gene cell death index model that can well predict the prognosis of ESCC patients by incorporating immunogenic cell death and necrosis features (*Cao et al., 2024*). In this study, we obtained genes related to 12 types of PCD from previous studies and identified five key PCD effectors by differential expression analysis and three machine learning analyses (linear SVM, RandomForest and LASSO analysis). The roles of these genes as diagnostic variables of ESCC and the potential mechanism of mediating their effects on ESCC were comprehensively analyzed.

Computer-assisted detection has been applied to identify early gastrointestinal lesions (*Hussein et al., 2022*). Machine learning algorithms are the primary approach to computer data analysis and predictive modeling (*Freitas et al., 2023*), and their use in clinical image analysis and interpretation could provide valuable information for early detection of ESCC (*Hosseini et al., 2023*). Although machine learning has been applied to predict clinical outcomes in various environments (*Fazzari et al., 2022*), detection of ESCC depends largely on the accuracy of the algorithm used by the type and data quality for training (*Hosseini et al., 2023*). In this study, *INHBA*, *LRRK2*, *HSP90AA1*, *HSPB8*, *EIF2AK2* were identified by all three machine learning methods as hub genes of ESCC. More than 10 years ago, *Seder et al. (2009)* found that *INHBA* is overexpressed in ESCC and inhibition of its expression

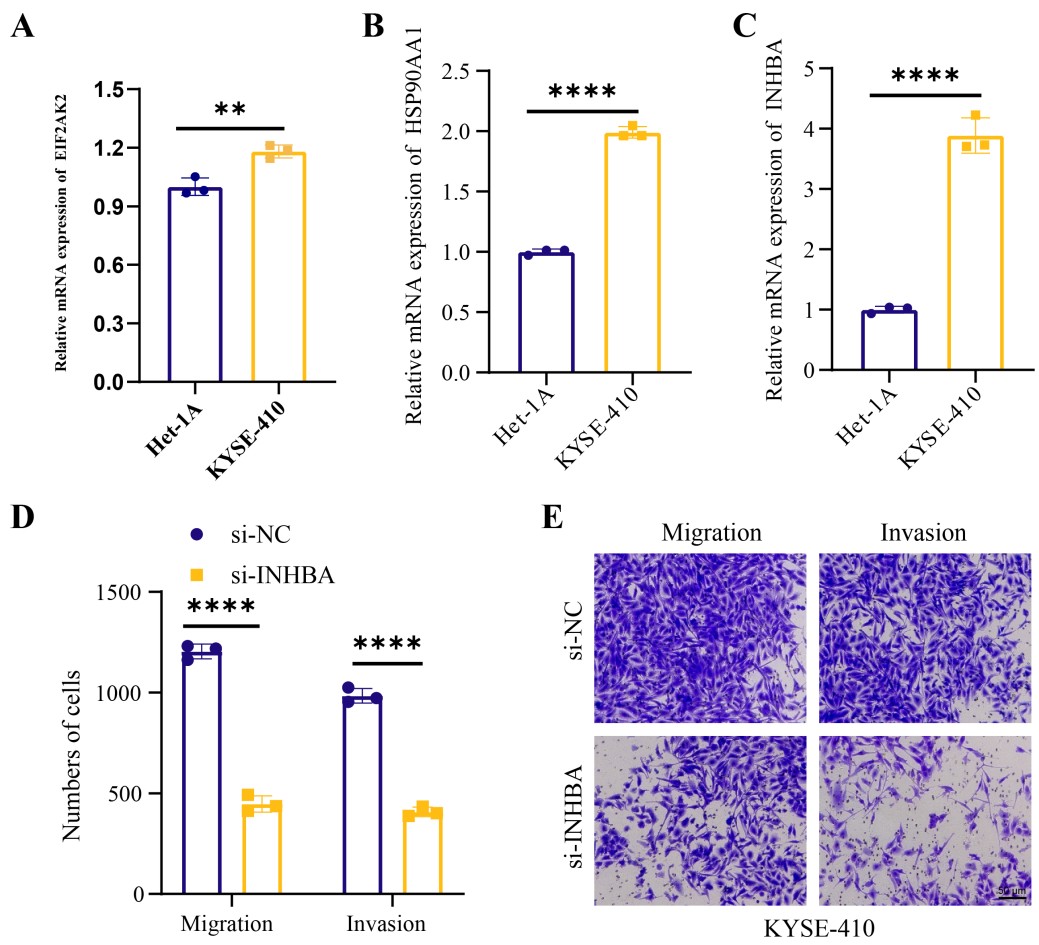

**Figure 6  Hub genes mediated the metastasis of ESCC cells.** (A–C) Relative mRNA levels of *EIF2AK2*, *HSP90AA1*, and *INHB* in KYSE-410 and Het-1A, respectively. (D–E) Effect of knockdown of *INHBA* on migration and invasion of KYSE-410 cells. Student's t-test was used to compare the differences in gene expression levels between the two groups. ** represents $p < 0.01$, **** represents $p < 0.0001$. Data were presented as mean + SEM.

suppresses the proliferative ability of cells. Furthermore, *Lyu et al. (2018)* showed that high expression of the *INHBA* gene is significantly associated with lymph node metastasis and poor prognosis in ESCC patients, therefore its overexpression is considered as a useful predictor. *LRRK2* is an autophagy-related protein kinase, and inhibition of *LRRK2* kinase activity stimulates macroautophagy (*Manzoni et al., 2013*). *LRRK2* has been reported to be associated with the risk of Crohn's disease in gastrointestinal disorders (*Foerster et al., 2022*). A pan-cancer analysis of *LRRK2* revealed that *LRRK2* increases the risk of low-grade glioma but serves as a protective factor for survival of patients with cutaneous melanoma (*Yan et al., 2022*). *HSP90AA1* is also an important regulator of autophagy that promotes autophagy by mediating the PI3K/Akt/mTOR cascade and inhibits cell apoptosis through the JNK/P38 pathway in osteosarcoma (*Xiao et al., 2018*). *HSP90AA1* is a carcinogenic enhancer in ESCC, and inhibition of its activity significantly weakens cell

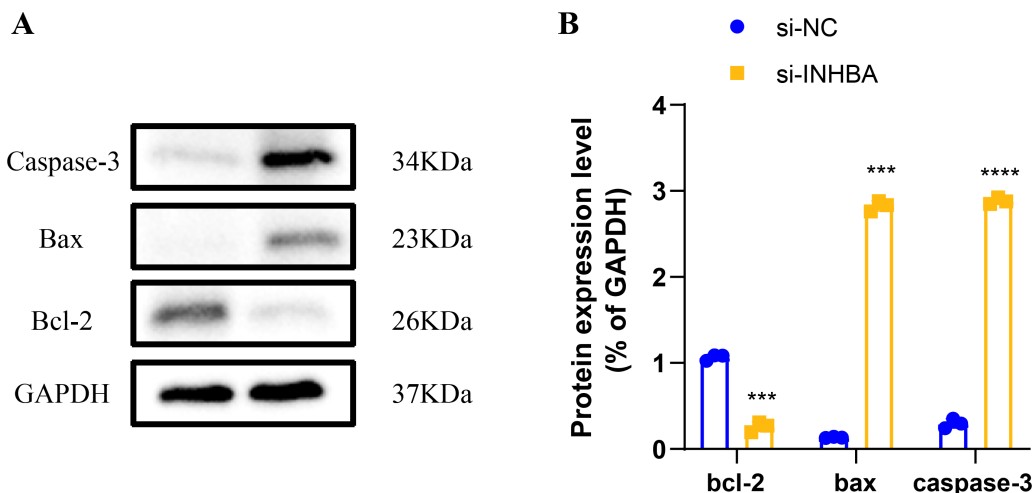

**Figure 7** **Modulation of apoptotic protein expression by *INHBA* silencing in ESCC cells.** (A) Western blotting-based assay to determine the expression levels of apoptosis marker proteins Caspase-3, Bax, and Bcl-2 after *INHBA* silencing. (B) Quantitative analysis of apoptotic protein expression changes upon *IN-HBA* silencing. *** represents $p < 0.001$, **** represents $p < 0.0001$. Data were presented as mean + SEM.

proliferation and induces cell apoptosis, with a lower *HSP90AA1* expression predicting a better prognosis of ESCC (*Ye et al., 2021*). *HSPB8* also is a small chaperone involved in chaperone-assisted selective autophagy (*Cristofani et al., 2021*). *Cristofani et al. (2022)* have shown that HSPB8-induced autophagy is a key event in the elimination of melanoma cell growth. ProteinkinaseR encoded by *EIF2AK2* gene is involved in autophagy and cell pyroptosis induced by inflammatory bodies in nasopharyngeal carcinoma (*Shen et al., 2012*; *Jiang et al., 2020*). In this study, we found the mRNA levels of *EIF2AK2*, *HSP90AA1* and *INHBA* were significantly overexpressed in ESCC cells, and that knockdown of *INHBA* expression suppressed ESCC cell migration and invasion. Moreover, all the five PCD-related genes had ROC values above 0.97 for the diagnosis of ESCC, particularly, the ROC of the 5-gene model even reached 0.997.

In addition, each gene also affected the infiltration of different immune cells in ESCC. For example, *INHBA* is negatively correlated with B-cell infiltration and positively with macrophages, neutrophils and dendritic cells in gastric cancer (*Liu et al., 2023*). *KRRK2* is positively associated with macrophage recruitment in the immune microenvironment of pancreatic cancer and its deficiency impairs macrophage function (*Yan et al., 2022*). In addition, thermotherapy can promote macrophage M1 polarization in the immune microenvironment of triple-negative breast cancer tumors through exosome-mediated *HSPB8* transfer (*Xu et al., 2023*). These studies were consistent with our results that these five genes played an important role in regulating the tumor immune microenvironment, especially in macrophage activation and recruitment, in ESCC.

Post-transcriptional regulation of gene expression is performed by miRNAs, whereas TFs play a key role in transcriptional rate activation or repression in the pre-transcriptional phase (*Shaik et al., 2022*). We identified the upstream regulators of five PCD-related

genes, which were regulated by 22 miRNAs and 105 TFs. Among them, the interaction of miR-150-3p with *INHBA* has been reported to inhibit the proinflammatory polarization of alveolar macrophages in sepsis (*Liang et al., 2023*). *Ba et al. (2021)* found that miR-9-5p binds to the 3′ UTR of *HSP90AA1* mRNA and that miR-9-5p decreased *HSP90AA1* protein expression. Furthermore, they indicated that knockdown of ZNRD1-AS inhibited gastric cancer cell proliferation and metastasis by targeting the miR-9-5p/*HSP90AA1* axis. The regulation of these molecular mechanisms of the five PCD-related genes in ESCC still needs further investigation. However, there were some limitations in this study. First, we used data from a specific sample set, which may limit the generalizability of the results. AI cannot replace the clinician's understanding of the specific condition of patients and it cannot fully replace biochemical validation in the laboratory. In particular, our study is mainly based on bioinformatics tools to predict TFs and miRNAs. Therefore, further experimental validation is needed to determine the accuracy of these results. Finally, the use of specific machine learning models may lead to biased results, therefore our results should be verified using multiple algorithms and cross-validation.

## CONCLUSION

In general, this study successfully identified five ESCC-related key genes through the integrated application of differential expression analysis and machine learning algorithms. The 5-gene model showed high specificity and sensitivity in ESCC diagnosis. These findings provided new perspectives for understanding the molecular mechanisms of ESCC, but further studies are needed to validate the specific roles of these genes in disease progression and to strengthen the statistical reliability with larger sample size.

### Abbreviations

| | |
|---|---|
| **PCD** | Programmed cell death |
| **ESCC** | Esophageal squamous cell carcinoma |
| **ROC** | Receiver operating characteristic |
| **AUC** | Area under the curve |
| **GEO** | Gene Expression Omnibus |
| **DEG** | Differentially expressed gene |
| **BP** | Biological process |
| **CC** | Cell component |
| **MF** | Molecular function |
| **SVM** | Support vector machine |
| **LASSO** | Least Absolute Shrinkage and Selection Operator |
| **CART** | Classification and regression trees |
| **OOB** | Outofbag |
| **TF** | Transcription factor |

### Funding

This study was supported by Porphyromonas gingivalis promotes epithelial mesenchymal transition in esophageal squamous cell carcinoma by activating AKT/$\beta$- Catenin signaling pathway (No. 81972571). The funders had no role in study design, data collection and analysis, decision to publish, or preparation of the manuscript.

### Grant Disclosures

The following grant information was disclosed by the authors:
Porphyromonas gingivalis promotes epithelial mesenchymal transition in esophageal squamous cell carcinoma by activating AKT/$\beta$- Catenin signaling pathway:  81972571.

### Competing Interests

The authors declare there are no competing interests.

### Author Contributions

- Min Chen performed the experiments, analyzed the data, authored or reviewed drafts of the article, and approved the final draft.
- Yijun Qi conceived and designed the experiments, analyzed the data, prepared figures and/or tables, and approved the final draft.
- Shenghua Zhang conceived and designed the experiments, prepared figures and/or tables, and approved the final draft.
- Yubo Du performed the experiments, analyzed the data, prepared figures and/or tables, and approved the final draft.
- Haodong Cheng conceived and designed the experiments, authored or reviewed drafts of the article, and approved the final draft.
- Shegan Gao performed the experiments, authored or reviewed drafts of the article, and approved the final draft.

### Data Availability

Data are publicly available at GenBank (GSE53625, GSE43732) and The Cancer Genome Atlas Program (TCGA) (TCGA-ESCA (https://portal.gdc.cancer.gov/).

Raw data are available in GitHub and Zenodo: https://github.com/1MinChen/My-Raw-data.git.

1MinChen. (2024). 1MinChen/My-Raw-data: My data (v1.1.0). Zenodo. https://doi.org/10.5281/zenodo.11044082.

### Supplemental Information

Supplemental information for this article can be found online at http://dx.doi.org/10.7717/peerj.17690#supplemental-information.

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
