# Peer review of "Molecular insights into programmed cell death in esophageal squamous cell carcinoma"

_PeerJ, doi:10.7717/peerj.17690_

## Round 0.1 · original submission · Major Revisions

Please address the concerns of all reviewers and amend the manuscript accordingly.

Reviewer 1 ·

Basic reporting

no comment

Experimental design

no comment

Validity of the findings

no comment

Annotated reviews are not available for download in order to protect the identity of reviewers who chose to remain anonymous.

·

Basic reporting

English used throughout the Ms is acceptable and understandable.
Literature refernces are adequate.
Figures are clearly described.
However, overall the data which are shown in the Ms are not justifying the conclusions drawn.
Comments: please provide an explanation of abbreviations at their first mentioning.
Abstract: needs restructuring
Methods. knockdown of INHBA is not explained
It would be good to provide a Table with relevant genes which are characteristic for the different detah pathways.

Experimental design

aims of the Ms are described clearly. The authors wanted to identify biomarkers for ESCC for diagnsois and therapy predominatly by AI based methods.
The AI based methods are described adequately.
Ethical issues are addressed.
The description of methods could be improved. Especially the wet lab methods lack important informations.
Line 155: culture conditions need to be explained.
Line 168: Culture Medium and culture flasks need to be mentioned as well as the supplements.
line 231: immune corerlation data need to be confirmed by wet lab data such as immunophenotyping of patent blood. or it should be made clear what the limitaiosn of the used method is.
Line 256: to base the statements made by the authors on the migratory and invasive capacity of ESCC on exact 2 cell lines is an overinterpretation .

Validity of the findings

The data are for sure clinically relevant. Howvere, the conclusions which are drawn are an overinterpretation.
Conclusions are not justified on the bais of the data provided in the Ms. It is clear that AI can give hints with respect to pathways which are differentially used in cancer versus normal cells. Howvere the findings need to be confirmed by experimental work.
line 85: "AI ofefrs opportunity to enhance dual multi-target discovery for diagnsotic and treatment of ESCC" This sentence needs to be corrected and it should be made clear what AI can provide and what not.

Additional comments

The pros and cons of the AI methods need tom be stated clearly.
The limitaions of the AI based findings need to be outlined.

·

Basic reporting

In the submitted manuscript authors tried to find new programmed cell death-related diagnostic biomarkers for esophageal squamous cell carcinoma (ESCC) and deduce molecular processes in which they participate, using bioinformatic re-analyses of someone else' datasets and tested the impact of one of them on migration and invasion of one ESCC cell line.
Although 'Introduction' contains sufficient clinical background on ESCC, not a single gene and molecular process related to ESCC development and progression have been mentioned.
English writing is of satisfactory quality, however, there are few repetitions which should be shortened or omitted:
- Those two sentences on GSE53625 dataset in lines 96-98 and 103-104 should be merged.
- Sentences in lines 207-210 and 212-214 are mere repetitions.
There are also some ambiguities or errors which should be clarified or corrected:
- In 'Abstract' authors wrote: "while LRRK2 and HSPB8 were lacking in cancer tissues of ESCC" what is wrong since in Figure 3C their expression maybe is lower in cancer compared to normal tissue, but is i not negligible!
- In line 38, "ESCC cells" must be replaced with "ESCC cell line KYSE-410" since only one ESCC cell line was used! Also, in line 257 there should be either "ESCC cells KYSE-410" or "ESCC cell line KYSE-410".
- In line 271 authors wrote "we downloaded PCD related genes" so it is first unclear how genes could be downloaded, and second, in lines 91-92 it was written "Programmed cell death (PCD) -related genes in 12 models were obtained from the article published by Zou et al (15)"?!

Experimental design

Study behind this manuscript is to a great deal irreproducible since methods' descriptions lack many important information:
- for ALL used datasests, original reference must be cited and precisely stated which data from dataset was used, since, e.g., GSE53625 contains both mRNA and lncRNA expression data
- for ALL software and R-packages, used version number must be provided and proper reference cited (if published in scientific journal)
- for ALL used on-line tools and databases, valid URL must be provided and proper reference cited
- for ALL used instruments, precise model and vendor must be provided
- sequences (or at least catalog numbers) of siRNAs must be provided and transfection precisely described (number of seeded cells, concentration of transfected siRNAs, which transfection reagent, etc.)
- for qPCR, cycling conditions must be provided and explained how relative expression calculated, and if 2^-ddCt method was used DOI: 10.1006/meth.2001.1262 must be cited
- it is unclear which statistical test was used for testing normality of data distribution, and if also paired t-test was used since paired non-parametric test was used
- for all bar graphs (Figures 3C, 5C, 6A-D) it must be stated how data was presented, i.e., which measures of central tendency and data dispersion were presented on graphs
- for all graphs (e.g., Venn diagram) it must be clear how they where created (which software/package used)
- in figure legends it must be clearly stated what was the source of data presented on figures (e.g., GSE53625)

Validity of the findings

Validity of the findings are extremely low because authors did not use another independent data for validation (e.g., TCGA-ESCA dataset), did not try to asses prognostic or even predictive value of those 5 hub genes, associated expression of hub genes with other clinicopathological variables, while in vitro finding is in essence just a repetition what has been already greatly known ("As early as more than 10 years ago, Christopher at al. found that INHBA was overexpressed in esophageal adenocarcinomas, and inhibition of its expression suppressed the proliferative ability of cells (31)."). Related to that, rational behind silencing just INHBA gene was not given, while also some other cell-based in vitro assays could be performed to provide more new and original molecular mechanistic and functional insights, since now, for example, it seems that both immune infiltration analysis and molecular regulatory network analysis of hub genes were done just perfunctory! Former was not even mentioned in 'Discussion', while later was not properly addressed as it should it be. Also, it is not even clear if the expression of target genes of those miRNAs and transcription factors was analyzed within GSE53625 dataset?!

Additional comments

Human gene symbols must be written in italics.

---

## Round 0.2 · Major Revisions

Please address the remaining concerns of reviewer #2 and revise the manuscript accordingly.

Reviewer 1 ·

Basic reporting

no comment

Experimental design

no comment

Validity of the findings

no comment

Additional comments

no comment

Annotated reviews are not available for download in order to protect the identity of reviewers who chose to remain anonymous.

·

Basic reporting

Authors have substantially improved quality of this manuscript through revision, however, there are still many drawbacks which must be corrected or further improved.
There are still few unclear phrases:
Line 192: "All bioinformatics analysis data analyzed"
Line 197: "the default p value < 0.05 was a statistical significance." [was considered statistically significant]
Lines 230-231: "while the level of LRRK2 and HSPB8 in ESCC tissues was remarkably [what?] as compared to the normal tissues"
Statement "Genes with significantly high expression in cancer tissues were easier to be detected." (line 233-234) is logical and obvious and thus redundant.

Experimental design

Methods are still not properly described:
- for all used software and R-packages, used version number MUST be provided
- for GENODE database both URL and reference must be provided, for HTFtarget and TargetScan references, and for CIBERSORT URL
- for ALL used antibodies, catalog number and used dilution must be stated
- microscope used for transwell assay must be precisely described (type, model, vendor), while for Figure 6E either scale bar or magnification must be provided
- test for testing normality of data distribution must be stated in 'Statistical analysis' section
- statement "Measurement data were expressed as mean ± standard deviation" (lines 193-194) is not true, since, e.g., Figures 3C, 5C and S1A simply do not show that, while authors responded that they have presented data as mean ± SEM!
Therefore, in each figure legend it must be precisely stated how data were presented, and explained what those asterisks and "ns" mean.
- how results from the TCGA-ESCA dataset were obtained must be explained in detail in 'Materials and methods' section: DOI: 10.1038/nature20805 must be cited; data acquisition, preprocessing, DEG and survival analyses explained; also explained how optimal cutoff value was determined; which actual type of survival (overall, RFS, etc.) was analyzed, etc.
- the source of primers' and siRNA sequence must be stated (e.g., from certain paper, self-designed (and how, i.e., which software), designed by a company, etc.)
- the sequence or catalog number of si-NC must be also provided

Validity of the findings

From 'Molecular regulatory network analysis of the hub genes' section it must be clear if those are just putative TFs and miRNAs or experimentally confirmed.
Authors should carefully read PMID: 23638278 and accordingly/properly interpret results presented on Figures 5B and 5D, since for instance, all correlations presented on Figure 5D are in essence either week or negligible!
Similarly, results presented on Figure S1B were not even (properly) commented.

Additional comments

TCGA-ESCA dataset must be mentioned in "Availability of data and materials" statement.
Most gene symbols in figure legends are not in italics.
Supplementary figure legends are to scarce, i.e., uninformative.

---

## Round 0.3 · Minor Revisions

Please address remaining concerns of the reviewer and amend manuscript accordingly.

·

Basic reporting

Professional English is used throughout the whole article.
no comment

Experimental design

no comment

Validity of the findings

Conclusion is missing

Additional comments

the revised version has improved the conetnt of the Ms.

·

Basic reporting

No comment.

Experimental design

No comment.

Validity of the findings

No comment.

Additional comments

Authors have further improved quality of this manuscript but there is still some thing which they haven't implemented:
1) Sentence "The normality of the data distribution was assessed using the Shapiro-Wilk test." must come before the sentence "Student’s t-test and Wilcoxon rank sum test were used to compare differences...".
2) For Figure 6E either scale bar or magnification must be provided.
3) Also for PRIMARY antibodies catalog number and used dilution must be provided.
4) Since Figures 3C, 5C and S1A definitively do not present data with mean ± SEM, presumably median and interquartile range, it would be useful to in each figure legend precisely explain how data were presented (not obtained, AND THIS HAS NOTHING WITH "repeating the detailed experimental methods in each figure legend")!

---

## Round 0.4 · accepted · Accept

Since all the remaining concerns of the reviewers were adequately addressed, revised manuscript is acceptable now.